



# Intercomparison of GEOS-Chem and CAM-chem tropospheric oxidant chemistry within the Community Earth System Model version 2 (CESM2)

Haipeng Lin[1], Louisa K. Emmons[2], Elizabeth W. Lundgren[1], Laura Hyesung Yang[1], Xu Feng[1], Ruijun Dang[1], Shixian Zhai[1,3], Yunxiao Tang[4], Makoto M. Kelp[4,5], Nadia K. Colombi[4], Sebastian D. Eastham[6,7], Thibaud M. Fritz[7], Daniel J. Jacob[1,4]

[1]John A. Paulson School of Engineering and Applied Sciences, Harvard University, Cambridge, MA, USA
[2]Atmospheric Chemistry Observations and Modeling Laboratory, National Center for Atmospheric Research, Boulder, CO, USA
[3]Earth and Environmental Sciences Programme and Graduate Division of Earth and Atmospheric Sciences, Faculty of Science, The Chinese University of Hong Kong, Sha Tin, Hong Kong SAR, China
[4]Department of Earth and Planetary Sciences, Harvard University, Cambridge, MA, USA
[5]Department of Earth System Science, Stanford University, Stanford, CA, USA
[6]Joint Program on the Science and Policy of Global Change, Center for Global Change Science, Massachusetts Institute of Technology, Cambridge, MA, USA
[7]Laboratory for Aviation and the Environment, Department of Aeronautics and Astronautics, Massachusetts Institute of Technology, Cambridge, MA, USA

*Correspondence to*: Haipeng Lin (hplin@seas.harvard.edu)

**Abstract.** Tropospheric ozone is a major air pollutant and greenhouse gas. It is also the primary precursor of OH, the main tropospheric oxidant. Global atmospheric chemistry models show large differences in their simulations of tropospheric ozone budgets. Here we implement the widely used GEOS-Chem atmospheric chemistry module as an alternative to CAM-chem within the Community Earth System Model version 2 (CESM2). We compare the resulting simulations of tropospheric ozone and related species to observations from ozonesondes, the ATom-1 aircraft campaign over the Pacific and Atlantic, and the KORUS-AQ aircraft campaign over the Seoul Metropolitan Area. We find that GEOS-Chem and CAM-chem within CESM2 have similar tropospheric ozone budgets and concentrations usually within 5 ppb but important differences in the underlying processes including (1) photolysis scheme (no aerosol effects in CAM-chem), (2) aerosol nitrate photolysis, (3) $N_2O_5$ cloud uptake, (4) tropospheric halogen chemistry, and (5) ozone deposition to the oceans. Global tropospheric OH concentrations are the same in both models but there are large regional differences reflecting the above processes. Carbon monoxide is lower in CAM-chem (and lower than observations) because of higher OH concentrations in the northern hemisphere and insufficient production from isoprene oxidation in the southern hemisphere. CESM2 does not scavenge water-soluble gases in convective updrafts leading to some upper tropospheric biases. Comparison to KORUS-AQ observations shows successful simulation of oxidants under polluted conditions in both models but suggests insufficient boundary layer mixing in CESM2. The implementation and evaluation of GEOS-Chem in CESM2 contributes to the MUSICA vision of modularizing tropospheric chemistry in Earth system models.



## 1 Introduction

Ozone is a central species in atmospheric chemistry. It is a major air pollutant and greenhouse gas, and the primary source of the hydroxyl radical (OH) which is the main tropospheric oxidant (Monks et al., 2015). It is produced within the troposphere by nonlinear photooxidation processes involving hydrogen oxide radicals ($HO_x \equiv OH + $ peroxy), nitrogen oxide radicals ($NO_x \equiv NO + NO_2$), volatile organic compounds (VOCs), and ozone itself. It is extensively observed from surface sites, aircraft, sondes, and satellites, and is thus an important indicator of skill for chemical transport models (Hu et al., 2017). At the same time, comparisons with observations can be successful for the wrong reasons. Extensive intercomparisons of global models often show similar tropospheric ozone burdens but large differences in chemical source and sink magnitudes (Wu et al., 2007; Young et al., 2018), implying large differences in sensitivity to perturbations. This is a particular problem for chemistry-climate models that aim to quantify chemical feedbacks on climate change.

Here we compare two state-of-science atmospheric chemistry modules, GEOS-Chem and CAM-chem, within the Community Earth System Model (CESM2) (Danabasoglu et al., 2020). CAM-chem is the resident atmospheric chemistry module in CESM2 (Lamarque et al., 2012; Tilmes et al., 2015, 2016; Emmons et al., 2020). GEOS-Chem is widely used as an offline chemical transport model (CTM) driven by external meteorological data (Bey et al., 2001). GEOS-Chem is grid-independent and modularized, so that the chemical module describing local operations in 1-D model columns (including emissions, chemistry, and deposition) is separated from the transport module (Long et al., 2015). This allows independent implementation of the GEOS-Chem chemical module in other models, including online applications where chemistry is coupled to transport (Hu et al., 2018; Lin et al., 2020; Lu et al., 2020; Keller et al., 2021). The GEOS-Chem chemical module has been previously coupled to the WRF and GEOS meteorological models to investigate aerosol-chemistry-climate feedbacks (Feng et al., 2021; Moch et al., 2022). The same scientific code base is used in the offline CTM such that version updates developed for the CTM can be seamlessly passed on to the online applications.

Fritz et al. (2022) implemented the GEOS-Chem chemical module in CESM2 as the first application of that module to an open-source Earth system model (ESM) for community use. GEOS-Chem offers an alternative representation of atmospheric chemistry to CAM-chem within CESM2, contributing to the MUSICA (MUlti-Scale Infrastructure for Chemistry and Aerosols; Pfister et al., 2020) vision for CESM of allowing users to choose among a range of options for atmospheric chemistry. The GEOS-Chem emission component (HEMCO; Keller et al., 2014) has been previously implemented in MUSICA (Lin et al., 2021). Fritz et al. (2022) presented general comparisons between GEOS-Chem and CAM-chem in the CESM2 environment. They found good agreement between the two modules for stratospheric ozone, but 10-30% lower tropospheric ozone in GEOS-Chem due to bromine chemistry not considered in CAM-chem. They found several challenges in the implementation of the GEOS-Chem chemical module within CESM2. For example, CESM2 uses the MAM4 (Modal Aerosol Model version 4; Liu et al. 2016) modal aerosol microphysics to simulate aerosol-cloud interactions and aerosol-



radiation interactions, while GEOS-Chem uses either bulk or sectional representations of aerosol microphysics. CESM2 does not explicitly include scavenging of water-soluble species in convective updrafts other than through MAM4, but this is a major process in wet deposition in the GEOS-Chem CTM to prevent unphysical buildup of soluble species in the upper troposphere (Balkanski et al., 1993; Liu et al., 2001). Indeed, Fritz et al. (2022) found large overestimates of upper tropospheric aerosol in GEOS-Chem within CESM2 as compared to the offline GEOS-Chem.

Our work builds on the Fritz et al. (2022) initial implementation of GEOS-Chem in CESM2 to address the previous challenges and to give a more thorough evaluation with observations and intercomparison with CAM-chem. We focus on tropospheric ozone and related oxidant chemistry from both a global perspective (ozonesonde and ATom-1 aircraft observations) and polluted conditions over East Asia (KORUS-AQ aircraft observations). KORUS-AQ, conducted in May-June 2016, is of particular interest because of the previously identified large differences between offline GEOS-Chem and CAM-chem in simulating the aircraft observations including 20-30 ppb differences in ozone (Park et al., 2021). We analyze the individual processes driving differences between GEOS-Chem and CAM-chem, and use observations to arbitrate when possible. As part of resolving differences in photolysis rates we implement into CAM-chem the Fast-JX photolysis scheme used in GEOS-Chem (Bian and Prather, 2002), further contributing to the MUSICA vision of process-level modularization of atmospheric chemistry models.

## 2 Model description and methods

### 2.1 CESM2, CAM-chem, and HEMCO

We use a beta version of CESM 2.3 including the CAM6 Community Atmosphere Model (CAM tag version `cam6_3_095`), which has provided the basis for the integration of the GEOS-Chem module into the mainline CESM code. All simulations are for the year 2016 with an 18-month initialization period. The year was chosen for evaluation with the ATom (Wofsy et al., 2018) and KORUS-AQ (Crawford et al., 2021) aircraft campaigns. We use a global 0.9° × 1.25° grid with 32 vertical layers up to 2 hPa. The model is nudged to reanalysis meteorology (using the "`FCnudged`" configuration in CAM6) from 3-hourly MERRA2 data available from the NASA Global Modeling and Assimilation Office.

CAM-chem is the standard chemistry representation in CESM2 with tropospheric-stratospheric chemistry, currently using the MOZART-TS1 (Model for OZone And Related chemical Tracers; Emmons et al., 2020) mechanism and the Modal Aerosol Model with 4 modes (MAM4; Liu et al., 2016) as default. MOZART-TS1 includes 229 chemical species and 541 reactions. Photolysis is calculated using a lookup table based on the Tropospheric Ultraviolet and Visible (TUV) radiation model, which takes into account the impact of clouds but not aerosols (Kinnison et al., 2007). A sensitivity simulation developed for this project uses Fast-JX instead of the TUV look-up table for photolysis.



The CAM-chem version in our work uses HEMCO for emissions but is otherwise unmodified. HEMCO is the standard emission component of GEOS-Chem (Keller et al., 2014), now implemented in CESM as part of MUSICA (Lin et al., 2021). It allows the use of any emission inventories on any grid to be supplied to the model in netCDF format at runtime with options to add, supersede, and scale emissions. Here we use the same emissions in GEOS-Chem and CAM-chem processed

through HEMCO. This includes global anthropogenic emissions from the CEDSv2 inventory (Community Emissions Data System; McDuffie et al., 2021) superseded by the KORUSv5 inventory (Woo et al., 2020) over East Asia. Fire emissions are from the GFED4.1s inventory (van der Werf et al., 2017; Randerson et al., 2018). HEMCO has extensions to use emission modules dependent on environmental variables and this is applied to soil $NO_x$ emissions from Hudman et al. (2012) and ocean iodine emissions from Sherwen et al. (2016a, 2016b). We otherwise use emissions computed from other modules in

CESM to enforce consistency of the atmospheric chemistry simulation with other CESM components. This includes biogenic VOC emissions from MEGANv2.1 (Guenther et al., 2012) computed with the Community Land Model (CLM) and lightning $NO_x$, dust, and sea salt emissions from CAM (Price et al., 1997; Mahowald et al., 2006a, 2006b; Lamarque et al., 2012).

### 2.2 GEOS-Chem within CESM2

Unless explicitly written otherwise, GEOS-Chem in this work refers to the online implementation of the GEOS-Chem chemical module within the CESM2 model and not the offline CTM. We use GEOS-Chem version 14.1.1 (doi:10.5281/zenodo.7696632) with the addition of particulate nitrate ($pNO_3^-$) photolysis following Shah et al. (2023), which was subsequently implemented in version 14.2.0 (doi:10.5281/zenodo.8411433). The same GEOS-Chem chemical module and MERRA-2 meteorological fields are used in CESM2 and in the offline CTM simulations presented here. The GEOS-

Chem chemical mechanism has 286 species and 914 reactions with a development history independent of MOZART-TS1. It features recent major updates to $NO_x$ heterogeneous and cloud chemistry (Holmes et al., 2019), isoprene chemistry (Bates and Jacob, 2019), aromatic chemistry (Bates et al., 2021), and Cl-Br-I tropospheric halogen chemistry (Wang et al., 2021). Photolysis is calculated using the Fast-JX model (Bian and Prather, 2002) with consistent aerosol and overhead column ozone information from the GEOS-Chem simulation (Eastham et al., 2014). No aerosol microphysics is included here so that

aerosol concentrations are represented by the bulk masses of their chemical components (Park et al., 2004; Pai et al., 2020) but with four size bins for dust and two for sea salt aerosol (Alexander et al., 2005; Fairlie et al., 2010).

Fritz et al. (2022) describe the original implementation of GEOS-Chem within CESM2. They developed an interface to pass input data to GEOS-Chem, run the GEOS-Chem chemical module, and export the updated chemical species concentrations.

The interface converts between the bulk aerosols in GEOS-Chem and the modal aerosols in MAM4 for aerosol-radiation and aerosol-cloud interactions. Coupling of the GEOS-Chem chemical module to CESM2 required the adaptation of several components for compatibility with CESM2 or consistency with CAM-chem. We summarize in Table 1 the important



differences between the atmospheric chemistry representations in CAM-chem, GEOS-Chem within CESM2, and the offline GEOS-Chem CTM.


Here we make several improvements and corrections to the original implementation of GEOS-Chem within CESM2 by Fritz et al. (2022). We simulate nucleation in MAM4 by passing the gas-phase $H_2SO_4$ production rate computed in GEOS-Chem from the $SO_2 + OH$ reaction. We add an aerosol sink in the upper troposphere and lower stratosphere following Hodzic et al. (2015, 2016) to compensate for CESM2's omission of coupling convective transport and scavenging. We correct the sea

surface temperatures passed to HEMCO, which results in inorganic iodine emissions being 1% of the previous incorrectly calculated value. We also add numerous GEOS-Chem diagnostics for analyzing model output, including individual reaction rates and total production and loss rates for individual species.



**Table 1** Major differences between CAM-chem and GEOS-Chem simulations.

| Simulation | CAM-chem within CESM2 | GEOS-Chem within CESM2 | Offline GEOS-Chem CTM |
|---|---|---|---|
| Meteorology | CESM2.3 nudged to MERRA2 [a] | | MERRA2 |
| Chemistry mechanism | MOZART-TS1<br><br>229 species and 541 reactions<br><br>$O_x$-$NO_x$-VOC-aerosol | GEOS-Chem v14.1.1<br><br>286 species and 914 reactions<br><br>$O_x$-$NO_x$-VOC-halogen-aerosol | |
| Photolysis | TUV lookup table | Fast-JX | |
| Aerosol microphysics | MAM4 modal aerosols [b] | | Bulk aerosols [c] |
| Aerosol composition | Sulfate<br><br>SOA (five VBS bins)<br><br>Primary organic matter<br><br>Black carbon<br><br>Soil dust (three modes)<br><br>Sea salt (three modes) | Sulfate, Nitrate, Ammonium<br><br>SOA (four VBS bins [d])<br><br>Primary organic carbon<br><br>Black carbon<br><br>Soil dust (four size bins)<br><br>Sea salt (two size bins) | |
| Dry deposition velocities (over land) | Computed by CLM | | Computed by GEOS-Chem |
| Dry deposition velocities (over ocean and sea ice) | Computed by CAM | Computed by GEOS-Chem | |
| Wet deposition | Gases: Neu scheme [e]<br><br>Aerosols: MAM4 | Gases: Neu scheme [e]<br><br>Aerosols: scavenged as $HNO_3$ | GEOS-Chem wet deposition scheme [f] |
| Scavenging in convective updrafts | Not explicitly simulated (see Section 6) | | Explicitly simulated |
| Lightning $NO_x$ parameterization | Price et al. (1997); 2.8-3.0 Tg N a$^{-1}$ | | Murray et al. (2012);<br><br>5-6 Tg N a$^{-1}$ |

[a.] with 50-h relaxation time.

[b.] GEOS-Chem bulk aerosols masses are mapped to MAM4 modes for aerosol-radiation and aerosol-cloud interaction effects within CESM2. See Fritz et al. (2022) for the species mapping between GEOS-Chem species to MAM4 aerosols.

[c.] Sectional aerosol microphysics are available in GEOS-Chem (Yu and Luo, 2009; Kodros and Pierce, 2017) but are not used here.

[d.] SOA ≡ secondary organic aerosol, VBS ≡ volatility basis set. GEOS-Chem here uses the Complex SOA option from Pye et al. (2010).

[e.] Neu and Prather (2012).

[f.] Liu et al. (2001) for water-soluble aerosols and Amos et al. (2012) for gases.





## 3 Comparison of photolysis schemes: Fast-JX and TUV

Figure 1 shows the mean photolysis frequencies ($J$ values) for $NO_2$ ($J_{NO2}$) and $O_3$ to $O(^1D)$ ($J_{O1D}$) simulated by the GEOS-Chem model (with Fast-JX) and the difference with CAM-chem (with TUV lookup table) in surface air in July. Photolysis rates in GEOS-Chem with Fast-JX are generally lower than in CAM-chem with TUV. Differences for $J_{NO2}$ are typically 0-
10% over oceans and 10-20% over land, while differences for $J_{O1D}$ are typically 10-20% over oceans and 20-40% over land. There are some larger differences in polluted and open fire regions and at high latitudes.

Fast-JX and TUV use the same spectroscopic data from the NASA JPL recommendations (Burkholder et al., 2020). Fast-JX includes aerosol extinction but TUV does not, which explains the larger differences over polluted and open fire regions.
Differences over the oceans are mainly due to clouds. While Fast-JX and TUV both represent effects of cloud extinction, treatment of cloud scattering between the two schemes can be different. The effects of aerosol-cloud interactions on cloud properties through MAM4 cause GEOS-Chem and CAM-chem to have different cloud optical depths that can lead to further differences. Cloud effects are particularly large at high latitudes because of low Sun angles. Sensitivity simulations for clear sky (no cloud or aerosol extinction input to the photolysis schemes) show smaller differences between Fast-JX and TUV,
generally less than 10% for $J_{NO2}$ and less than 20% for $J_{O1D}$, while sensitivity simulations with Fast-JX implemented in CAM-chem show less than 5% differences for $J_{NO2}$ and $J_{O1D}$ everywhere compared to Fast-JX implemented in GEOS-Chem.

Figure 2 shows photolysis frequencies from the KORUS-AQ and ATom-1 campaigns derived from actinic flux measurements (Hall et al., 2018; Crawford et al., 2021), compared to the photolysis frequencies computed by Fast-JX and
TUV sampled along the aircraft flight tracks. $J_{NO2}$ values agree within 10% and there is no systematic bias relative to observations. Fast-JX values tend to be higher than TUV at high altitudes and this can be attributed to cloud effects as discussed above. $J_{O1D}$ values also show good agreement for ATom-1 but observed values for KORUS-AQ are much lower than for ATom-1 in the same season, which is captured by Fast-JX but not by TUV (which is 30% too high). We find that the overestimate of $J_{O1D}$ by TUV during KORUS-AQ is due in part to not accounting for aerosol extinction. Comparison of
clear-sky $J$ values shows that there is some additional unidentified factor causing TUV to be too high during KORUS-AQ and this disappears when Fast-JX is implemented in CAM-chem. In what follows the CAM-chem simulation uses the TUV lookup table but we will comment as appropriate on the effect of switching to Fast-JX.



**Figure 1** Mean photolysis frequencies for NO$_2$ ($J_{NO2}$) and O$_3$ to O($^1D$) ($J_{O1D}$) in surface air in July 2016. The left panels show the values computed by Fast-JX within GEOS-Chem. The right panels show the differences ($\Delta$) with the values computed by TUV within CAM-chem.





**Figure 2** Median vertical profiles of $J_{NO2}$ and $J_{O1D}$ from the KORUS-AQ aircraft campaign over the Seoul Metropolitan Area (SMA; 37–
37.6° N, 126.6–127.7° E) in May-June 2016 and the ATom-1 aircraft campaign over the Pacific and Atlantic Oceans in July-August 2016. Observations from in situ measurements of actinic fluxes (Hall et al., 2018) are compared to CESM2 values using the Fast-JX scheme within GEOS-Chem and the TUV scheme within CAM-chem. The model values are sampled along the aircraft flight tracks.



# 4 Global budgets and distributions of tropospheric oxidants

Table 2 shows global tropospheric ozone and OH budgets from GEOS-Chem and CAM-chem compared to the literature.

Ozone budgets from the two models and the multi-model mean in Young et al. (2018) are within 10% of each other. The larger chemical production and shorter chemical lifetime in GEOS-Chem are mainly due to photolysis of particulate nitrate (Shah et al., 2023), without which chemical production in GEOS-Chem decreases by 10% to 4902 Tg a$^{-1}$ and the tropospheric ozone burden decreases by 5% to 332 Tg. The lower dry deposition in GEOS-Chem reflects lower ozone deposition to the ocean (Pound et al., 2020). GEOS-Chem and CAM-chem have the same global OH concentrations, on the

high end of the range of values from the ACCMIP and CCMI model ensembles (Naik et al., 2013; Zhao et al., 2019). The lifetime of methylchloroform against loss to tropospheric OH is 5.4 and 5.3 years respectively in GEOS-Chem and CAM-chem, 15% lower than 6.3 $\pm$ 0.4 years inferred from observations (Prather et al., 2012).

Figure 3 shows the spatial distribution of annual mean OH concentrations simulated by GEOS-Chem and the difference with

CAM-chem. Despite having the same global mean OH concentrations, the two models have large regional differences. GEOS-Chem is up to 30% lower than CAM-chem over the continents, particularly over polluted regions, due to lower $J(O^1D)$ and possibly higher OH reactivity. Over the Amazon and Congo basins where NO$_x$ is low, isoprene does not titrate OH in GEOS-Chem due to recent updates in isoprene oxidation chemistry (Bates and Jacob, 2019).

Figure 4 shows annual mean surface and 500 hPa ozone and NO$_x$ concentrations simulated by GEOS-Chem and differences with CAM-chem. Ozone differences are generally smaller than 5 ppb, indicating a remarkable degree of agreement. The largest surface differences are at southern mid-latitudes due to slower ozone deposition to the ocean in GEOS-Chem. At 500 hPa, GEOS-Chem has lower ozone at high latitudes due to tropospheric halogen chemistry not represented in CAM-chem, while ozone and NO$_x$ are higher over the tropics mainly due to particulate nitrate photolysis. NO$_x$ is lower in GEOS-Chem at

high latitudes due to loss to halogen nitrates (Wang et al., 2021) and N$_2$O$_5$ uptake in clouds (Holmes et al., 2019), not included in CAM-chem.

Table 3 shows global budget terms in the troposphere from sensitivity simulations varying the most important differences between GEOS-Chem and CAM-chem. The largest controlling factors for tropospheric ozone differences between GEOS-

Chem and CAM-chem are nitrate photolysis and tropospheric halogen chemistry, which increase and decrease the tropospheric ozone budget by 5% and 4% respectively. Particulate nitrate photolysis is necessary in GEOS-Chem to correct a low NO$_x$ bias over the oceans in comparison to ATom data (Shah et al., 2023), but CAM-chem NO$_x$ is 4% higher than GEOS-Chem despite not having particulate nitrate photolysis. Inspection of Table 3 indicates that this reflects both the lack of tropospheric halogen chemistry in CAM-chem and the use of TUV for photolysis. Using Fast-JX for photolysis in CAM-





chem results in a 7% decrease in tropospheric $NO_x$, which we attribute tentatively to lower $J_{NO2}$ in surface air over continents (Figure 1).

Fritz et al. (2022) previously found tropospheric ozone in GEOS-Chem to be 30% lower than CAM-chem in the extratropics because of halogen chemistry, but iodine emissions in that simulation were 100-fold too high (Section 2.2). With corrected
iodine emissions we find only a 4% decrease of tropospheric ozone in GEOS-Chem due to tropospheric halogen chemistry, which is lower than the 11-19% effect previously reported in offline GEOS-Chem simulations (Sherwen et al., 2016b; Wang et al., 2021). We find that this is due to weaker wind speeds and lower sea surface temperatures in CESM2, resulting in weaker sea salt and gaseous iodine emissions.

**Table 2** Global budgets of tropospheric ozone and OH. [a]

| Budget terms | GEOS-Chem | CAM-chem | Previous literature [b] |
|---|---|---|---|
| Tropospheric ozone burden (Tg) | 350 | 342 | 340 (250-410) |
| $O_x$ chemical production (Tg a$^{-1}$) | 5395 | 5052 | 4900 (3800-6900) |
| $O_x$ chemical loss (Tg a$^{-1}$) | 4813 | 4465 | 4600 (3300-6600) |
| $O_x$ deposition (Tg a$^{-1}$) | 878 | 967 | |
|    Ozone dry deposition (Tg a$^{-1}$) | 749 | 826 | 1000 (700-1500) |
| $O_x$ STE (Tg a$^{-1}$) [c] | 341 | 380 | 500 (180-920) |
| $O_x$ Lifetime (days) | 23.0 | 23.7 | 22.3 (19.9-25.5) |
| Global OH ($10^6$ molecule cm$^{-3}$) [d] | 1.21 | 1.22 | 1.11 ± 0.16 |
|    N/S ratio | 1.22 | 1.26 | MMM: 1.28 ± 0.10; Obs.: 0.85 - 0.98 |
|    $\tau_{MCF}$ (a) | 5.4 | 5.3 | MMM: 5.7 ± 0.9; Obs.: 6.3 ± 0.4 |
| Stratospheric ozone burden (Tg) | 2744 | 2744 | |

[a.] Annual mean values for 2016 from GEOS-Chem and CAM-chem in CESM2. The troposphere is defined by $O_3$ < 150 ppb (Young et al., 2013). The budget is for the odd oxygen ($O_x$) family to account for rapid cycling between $O_x$ species: $O_x \equiv O_3 + O + O(^1D) + NO_2 + 2NO_3$ $+ HNO_3 +$ particulate nitrate $+ HNO_4 + 3N_2O_5 +$ organic nitrates $+$ Criegee intermediates $+ XO + HOX + XNO_2 + 2XNO_3 + 2OIO + 2I_2O_2$ $+ 3I_2O_3 + 4I_2O_4 + 2Cl_2O_2 + 2OClO$, where $X$ is Cl, Br, or I. CAM-chem does not include particulate nitrate or tropospheric halogen
species.

[b.] Means and ranges from Young et al. (2018) (33 models) for ozone and Naik et al. (2013) (16 ACCMIP models) for OH, for the year 2000. MMM: Multi-model mean. Obs.: Observation-derived estimates.

[c.] Stratosphere-Troposphere Exchange (STE) is estimated from the residual of mass balance between tropospheric chemical production and loss, $O_x$ deposition, and accumulation.

[d.] Global annual mean air-mass-weighted OH concentration in the troposphere.




**Table 3** Global tropospheric budget terms from different configurations of GEOS-Chem and CAM-chem in CESM2. [a]

|  | GEOS-Chem | | | | CAM-chem | |
|---|---|---|---|---|---|---|
| Simulation | Standard | No nitrate photolysis | No N₂O₅ cloud uptake | No halogen chemistry [b] | Standard | Fast-JX photolysis |
| Ozone burden (Tg) | 350 | 332 | 356 | 365 | 342 | 355 |
| $O_x$ chemical production (Tg a$^{-1}$) | 5395 | 4902 | 5473 | 5048 | 5052 | 5233 |
| $O_x$ chemical loss (Tg a$^{-1}$) | 4813 | 4425 | 4882 | 4542 | 4465 | 4469 |
| $O_x$ lifetime (days) | 23.0 | 24.7 | 23.0 | 25.1 | 23.7 | 24.1 |
| Global OH ($10^6$ molecule cm$^{-3}$) [c] | 1.21 | 1.06 | 1.20 | 1.32 | 1.22 | 1.22 |
| $NO_x$ burden (Gmol N) | 8.66 | 8.25 | 8.97 | 9.61 | 9.03 | 8.36 |

[a.] Refer to footnotes in Table 2. The Standard entries replicate those of Table 2.

[b.] Zeroing out reaction rates for halogen reactions in the troposphere.

[c.] Global annual mean air-mass-weighted OH concentration in the troposphere.

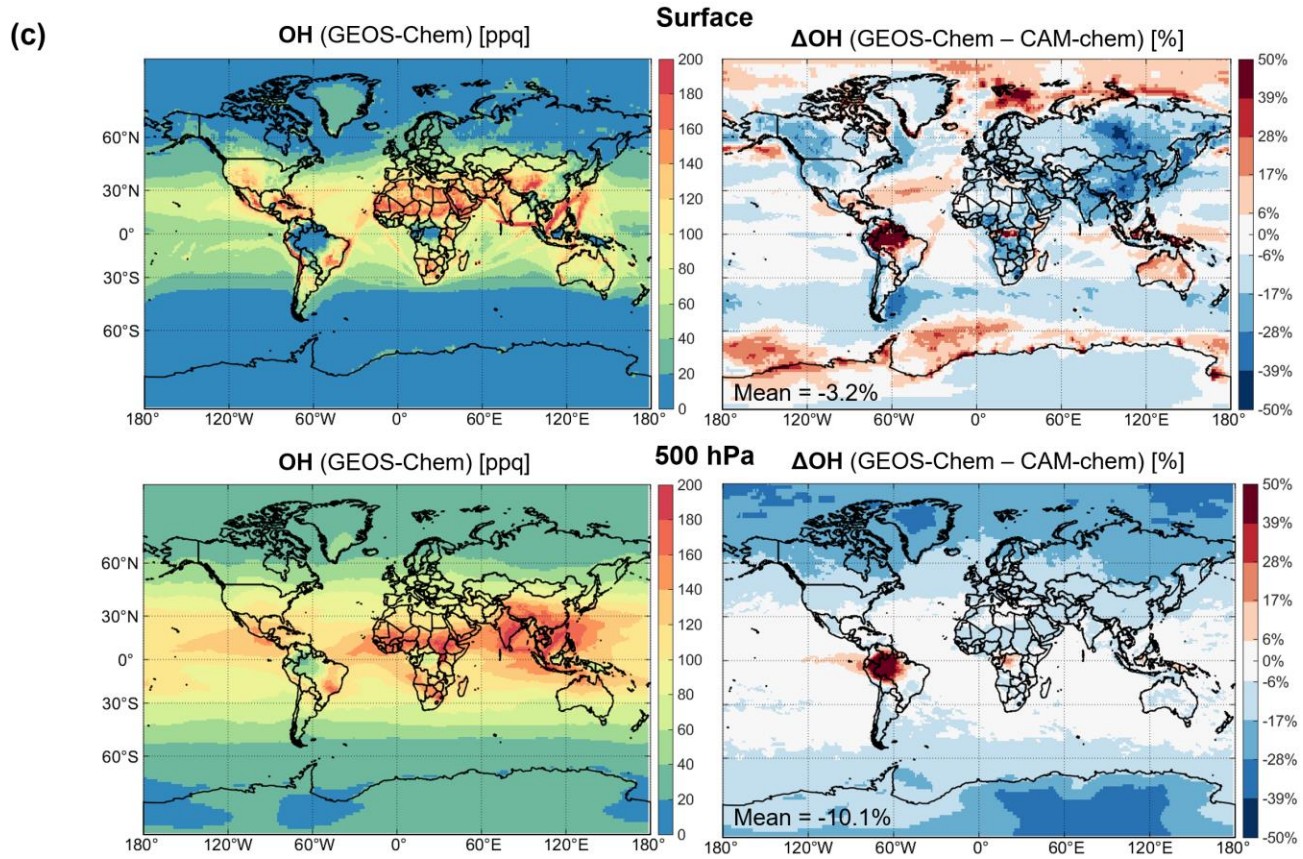

**Figure 3** Annual mean OH concentrations in surface air and at 500 hPa in GEOS-Chem within CESM2, and differences with CAM-chem. Percentage differences are relative to CAM-chem. Global mean differences are in legend. Values are for 2016.







**Figure 4** Annual mean ozone and NO$_x$ concentrations in surface air and at 500 hPa in GEOS-Chem within CESM2, and differences with CAM-chem. Percentage differences are relative to CAM-chem. Global mean differences are in legend. Values are for 2016.



**5 Comparisons to ozonesondes and to ATom-1 aircraft campaign**

Figure 5 compares annual mean ozone vertical profiles simulated by GEOS-Chem and CAM-chem to ozonesonde observations for 2016 from the World Ozone and Ultraviolet Radiation Data Centre (WOUDC), averaged across 9 regions
following Tilmes et al. (2012). Both models match the observations well and are within 5-10 ppb of each other. GEOS-Chem has lower ozone at high northern latitudes because of halogen chemistry.

Figure 6 shows tropospheric profiles of OH, NO, and CO simulated by GEOS-Chem and CAM-chem over the oceans in comparison to observations from the ATom-1 campaign. Both models generally agree with OH observations within
uncertainty. Both models fit NO observations within a factor of 2 in the northern hemisphere but have large underestimates in the southern hemisphere. This underestimate is a known model issue in previous offline GEOS-Chem simulations (Travis et al., 2020) and is not correctable by nitrate photolysis because particulate nitrate concentrations in the southern hemisphere are low (Shah et al., 2023). Observations show a NO increase in the upper troposphere of the southern hemisphere that is captured by CAM-chem but not GEOS-Chem. Previous work has shown that offline GEOS-Chem simulations capture this
increase of NO in the upper troposphere (Shah et al., 2023). A sensitivity GEOS-Chem simulation without tropospheric halogen chemistry, as shown in Figure 6, also captures this increase. One possible reason is CESM2 not explicitly representing scavenging in convective updrafts for soluble gases such as HBr and HOBr. This would increase the formation of stable halogen nitrates in the upper troposphere where thermolysis and hydrolysis are slow (Wang et al., 2021).

Both models underestimate CO in the northern hemisphere which is a known issue attributed to excessive OH (Gaubert et
al., 2020) or missing emissions of CO and its precursors (Park et al., 2021; Tang et al., 2023). CAM-chem has 10-20 ppb lower CO globally compared to GEOS-Chem that is likely driven by differences in OH. In the southern hemisphere the difference is driven by improvements in isoprene oxidation in GEOS-Chem by Bates and Jacob (2019), which recycles OH through H-shift isomerization of isoprene-hydroxy-peroxy radicals under low-NO conditions, seen in observations by Wells et al. (2020). This leads to faster in situ isoprene oxidation and a higher CO yield. This is not included in CAM-chem's
default MOZART-TS1 mechanism used in this work but is included in the updated MOZART-TS2 mechanism (Schwantes et al., 2020, 2021).





**Figure 5** Comparison of GEOS-Chem and CAM-chem simulated annual mean ozone vertical profiles to 2016 ozonesonde observations. The regions average a number *N* of observing sites as given by Tilmes et al. (2012). Horizontal bars are standard deviations of the means across the *N* sites.





**Figure 6** Median vertical profiles of OH, NO, and CO concentrations from the ATom-1 field campaign (July-August 2016) and from the GEOS-Chem and CAM-chem models within CESM2. Observations are separated between northern and southern hemispheres (NH and SH), filtered to remove influences from biomass burning ($CH_3CN > 200$ ppt; Travis et al., 2020) and binned in 1km intervals. Shaded areas correspond to the measurement accuracy.



## 6 Comparison to KORUS-AQ aircraft campaign

We use comparison to observations from the KORUS-AQ campaign (May 1 to June 10, 2016) over the Seoul Metropolitan Area (SMA, 37–37.6° N, 126.6–127.7° E) as illustrative of a polluted atmosphere. Figure 7 shows median concentration

profiles of oxidants and related species. Observations are compared to GEOS-Chem and CAM-chem sampled along the flight tracks, and to GEOS-Chem sensitivity simulations without particulate nitrate photolysis and without the nitrate correction applied in CESM2 for lack of scavenging in convective updrafts. Also shown in the Figure are vertical profiles from an offline regional GEOS-Chem simulation reported by Yang et al. (2023). Ozone vertical profiles in GEOS-Chem and CAM-chem are comparable and consistent with observations (Fig. 7a). Successful simulation of ozone in GEOS-Chem is

contingent on particulate nitrate photolysis, offsetting the loss from halogen chemistry (Colombi et al., 2023; Yang et al., 2023). A previous ozone multi-model intercomparison with KORUS-AQ observations by Park et al. (2021) found GEOS-Chem without particulate nitrate photolysis to be too low, consistent with the results shown here. CAM-chem was the only model to successfully reproduce observed ozone in that intercomparison and this was attributed to its stratospheric ozone influx, but here GEOS-Chem uses the same dynamics and hence the same stratospheric influx. The success of CAM-chem in

KORUS-AQ reflects instead its non-accounting of tropospheric halogen chemistry as a sink of ozone, which in GEOS-Chem needs to be compensated by particulate nitrate photolysis.

Particulate nitrate photolysis increases free tropospheric $NO_x$ and ozone production but this depends on the nitrate concentration. CAM-chem does not simulate nitrate. Because GEOS-Chem nitrate is not removed in convective updrafts in

the CESM2 environment, our standard implementation within CESM2 corrects nitrate using the same photolytic sink that CAM-chem applies for SOA with a rate of $0.0004 \times J_{NO2}$ and no products (Hodzic et al., 2015, 2016) to avoid buildup in the upper troposphere. Nitrate in GEOS-Chem within CESM2 without this correction (dotted line in Fig. 7c) is overestimated above 3km which leads to excessive effects of particulate nitrate photolysis including overestimation of ozone (Fig. 7a) and NO (Fig. 7b). The offline GEOS-Chem simulation by Yang et al. (2023) explicitly scavenges nitrate in convective updrafts

and does not overestimate particulate nitrate or NO. A solution would be to replace CESM convective transport with the GEOS-Chem offline convective transport and scavenging module using archived CESM convective mass fluxes, and this has been done before when coupling GEOS-Chem to the GEOS and BCC ESMs which had the same problem of not scavenging water-soluble species in convective updrafts (Yu et al., 2018; Lu et al., 2020). A more comprehensive solution would be to include scavenging of water-soluble species in the CESM2 convection scheme. This is implemented for MAM aerosols

(Wang et al., 2013) but not for gas-phase species or aerosols only represented in GEOS-Chem, including nitrate.

The simulations of particulate nitrate and peroxyacetylnitrate (PAN) within CESM show a sharp drop of concentrations with altitude above the surface, whereas the observations and the offline GEOS-Chem simulation of Yang et al. (2023) show a mixed layer structure extending to 1-2 km altitude. This likely reflects a bias in the CESM2 boundary layer mixing scheme



that would need to be investigated further. Boundary layer mixing in the offline GEOS-Chem model is a standard non-local
       scheme from J. Lin and McElroy (2014). The PAN simulations in GEOS-Chem and CAM-chem otherwise agree closely,
       indicating similar production from VOC chemistry, and are lower than the offline GEOS-Chem simulation which includes
       additional emissions of volatile chemical products (VCPs) as a source of acetaldehyde leading to PAN production (Yang et
       al., 2023).


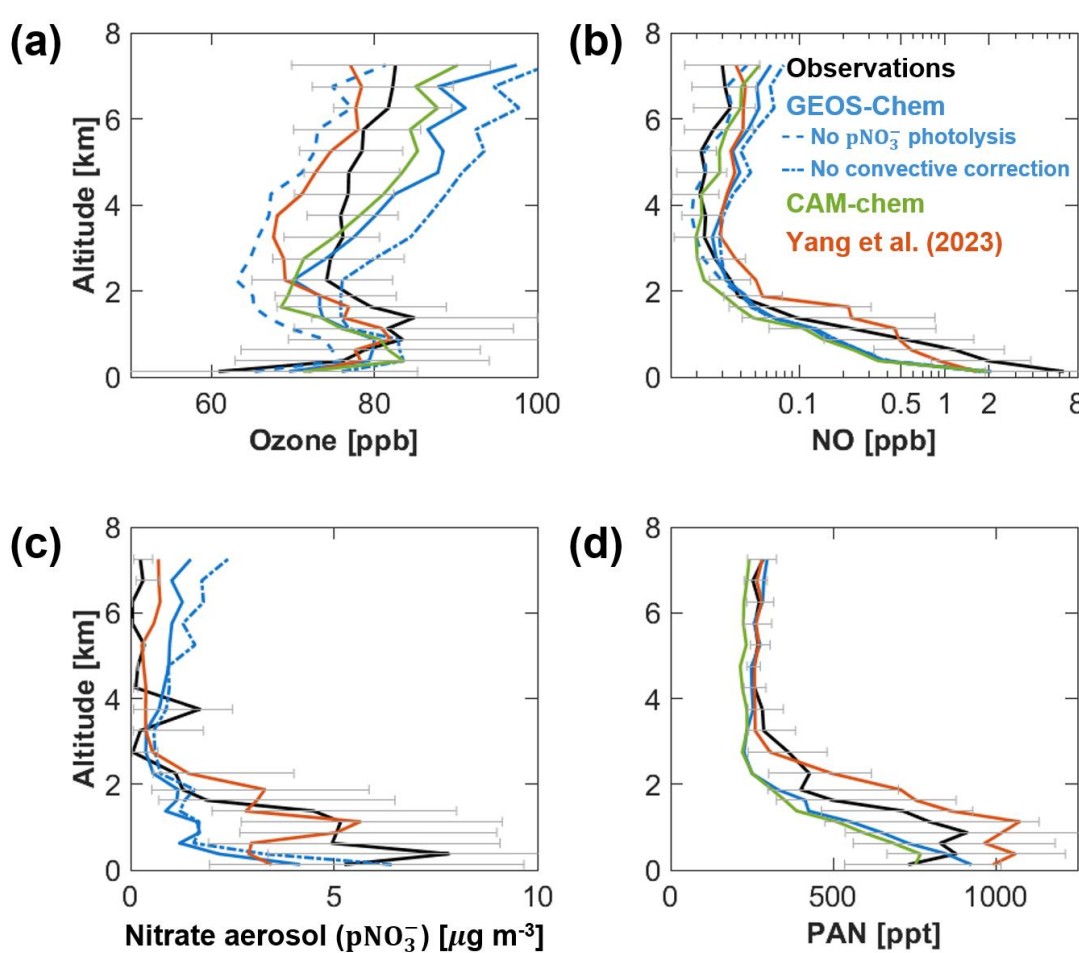

**Figure 7** Median tropospheric vertical profiles of species concentrations over the Seoul Metropolitan Area (SMA; 37–37.6° N, 126.6–127.7° E). during the KORUS-AQ aircraft campaign in May-June 2016. Observations are compared to GEOS-Chem and CAM-chem simulations within CESM2, and to the offline GEOS-Chem simulation reported by Yang et al. (2023). Results from a GEOS-Chem
sensitivity simulation with no particulate nitrate photolysis and no correction for scavenging of nitrate in wet convective updrafts are also shown. The vertical profiles are constructed by binning data into 0.25 km intervals below 2 km altitude and 0.5 km intervals above 2 km altitude. Horizontal bars represent the interquartile range of the observations in the given vertical bin.





## 7 Conclusions

GEOS-Chem has been implemented as an atmospheric chemistry module in the NCAR Community Earth System Model
(CESM2) to serve as alternative to CAM-chem and contribute to the MUSICA vision of plug-and-play modularization of
atmospheric chemistry within CESM (Pfister et al., 2020). Here we presented an intercomparison and evaluation with
observations of tropospheric oxidant simulations with these two modules. The intercomparison covered the full year of 2016,
allowing evaluation with the ATom-1 aircraft campaign over the remote Pacific and Atlantic, and the KORUS-AQ aircraft
campaign over the Seoul Metropolitan Area (SMA). Both GEOS-Chem and CAM-chem used the same emissions processed
through HEMCO (Lin et al., 2021) and the same coupling to other CESM2 modules.

GEOS-Chem uses the Fast-JX scheme for photolysis while CAM-chem uses a lookup table based on TUV. Both schemes
agree to within 10% when compared to $J_{NO2}$ and $J_{O1D}$ photolysis frequencies observations in ATom-1, but observations in
KORUS-AQ show that CAM-chem overestimates $J_{O1D}$ while GEOS-Chem dos not. One major difference is that TUV does
not account for extinction by aerosols while Fast-JX does. We implemented Fast-JX in CAM-chem and find that it resolves
most of the photolysis differences with GEOS-Chem.

Global tropospheric ozone budget terms in GEOS-Chem and CAM-chem agree within 10%, compared to a much wider
spread in the literature and due in part to canceling effects. Differences between the two models are mostly driven by aerosol
nitrate photolysis, $N_2O_5$ uptake in clouds, and tropospheric halogen chemistry, all of which are included in GEOS-Chem but
not in CAM-chem. Aerosol nitrate photolysis in GEOS-Chem produces $NO_x$ and enhances ozone production, compensating
for losses from $N_2O_5$ uptake in clouds and tropospheric halogen chemistry. Annual mean ozone concentrations agree within
5 ppb between GEOS-Chem and CAM-chem almost everywhere. Lower ozone deposition to the oceans in GEOS-Chem
results in higher surface ozone at southern mid-latitudes. Tropospheric halogen chemistry results in lower ozone at high
northern latitudes. Tropospheric $NO_x$ in GEOS-Chem is higher than CAM-chem in the tropics due to nitrate photolysis, and
lower at high latitudes due to $N_2O_5$ uptake by cloud and formation of halogen nitrates. The global mean tropospheric OH
concentration is identical between the two models but there are large differences over the continents driven by photolysis and
by isoprene chemistry.

Both GEOS-Chem and CAM-chem show good agreement with annual mean ozonesonde observations over the range of
latitudes. Comparison to ATom-1 observations in July-August 2016 shows good agreement for OH concentrations in both
the northern and southern hemispheres (NH and SH) within the measurement accuracy, and for $NO_x$ in the NH, but $NO_x$ in
the SH is underestimated. GEOS-Chem shows a depletion of $NO_x$ in the SH upper troposphere that is due to formation of
halogen nitrates and is not seen in the observations. However, the offline GEOS-Chem simulation does not show this
problem. One issue in CESM2 is the lack of scavenging of water-soluble species including halogen radical reservoirs in



convective updrafts. Both GEOS-Chem and CAM-chem underestimate CO in the NH but CAM-chem is consistently lower than GEOS-Chem due to higher OH in the NH and suppression of CO production from isoprene oxidation in the SH.

Comparison with KORUS-AQ aircraft observations allowed model evaluation for polluted conditions. Vertical ozone
profiles in GEOS-Chem and CAM-chem agree well with observations, which in GEOS-Chem is contingent on the $NO_x$ source from particulate nitrate photolysis. Lack of scavenging of GEOS-Chem aerosols such as nitrate and its gas phase precursors in convective updrafts is a major shortcoming in CESM2 that hinders proper representation of nitrate and other aerosols in the upper troposphere. Comparison of peroxyacetylnitrate (PAN) shows good agreement between GEOS-Chem and CAM-chem and with observations, indicating consistency in the VOC chemistry producing PAN. However, the decrease
of PAN and particulate nitrate mixing ratios with altitude are much sharper than observed and simulated by the offline GEOS-Chem model, implying insufficient boundary layer mixing in CESM2.

Overall, we have shown that GEOS-Chem provides a high-quality simulation of tropospheric oxidant chemistry in CESM2 and can contribute modules for alternative representations of atmospheric chemistry to serve the MUSICA vision.



*Code availability.* A fork of an alpha version (cam6_3_095) of the Community Atmosphere Model (CAM) including GEOS-Chem is available at https://github.com/CESM-GC/CAM and is used in this work. CAM-chem using HEMCO for emissions is implemented in the mainline CAM code as of cam6_3_118 (https://github.com/ESCOMP/CAM/tree/cam6_3_118). GEOS-Chem within CESM2 is implemented in the mainline CAM code as of cam6_3_147 (https://github.com/escomp/cam/tree/cam6_3_147).


*Author contributions.* LKE, SDE, and DJJ conceived of the project, acquired funding, and supervised the work. HL, EWL, XF, SDE, and TMF performed software development. HL, LKE, LHY, XF, RD, SZ, YT, MK, NKC, and DJJ analyzed the model results. SZ prepared the KORUSv5 emission inventory for input into HEMCO and CESM2. HL performed the visualization and preparation of the original draft. All authors contributed to review and editing of the manuscript.


*Competing interests.* The contact author has declared that none of the authors have any competing interests.

*Acknowledgments.* This work was supported by the Atmospheric Chemistry Program of the US National Science Foundation. We would like to acknowledge high-performance computing support from Cheyenne: HPE/SGI ICE XA

System (https://doi.org/10.5065/D6RX99HX; Hart, 2021) and Derecho: HPE Cray EX System (https://doi.org/10.5065/qx9a-pg09), provided by NCAR's Computational and Information Systems Laboratory (CISL) and sponsored by the National Science Foundation.



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
