# Peer review of "Intercomparison of GEOS-Chem and CAM-chem tropospheric oxidant chemistry within the Community Earth System Model version 2 (CESM2)"

_EGUsphere, 2024_

## Author Response (AR1)

**Response from authors**

Re: Reviews of "Intercomparison of GEOS-Chem and CAM-chem tropospheric oxidant

chemistry within the Community Earth System Model version 2 (CESM2)"

June 20, 2024

We thank the three Reviewers for their helpful comments. In response to their feedback, we have carefully revised the manuscript to address their concerns and clarify the differences between the two models in our work.

We respond to each specific comment in detail below. Referee comments are shown in red italics. Our replies are shown in black and modified text is shown in blue. Annotated page numbers refer to the revised copy of the manuscript. A tracked-changes copy of the manuscript is uploaded to the response materials.

Sincerely,

Haipeng Lin on behalf of the co-authors.

**Reviewer #1**

*This study compares tropospheric ozone simulations using GEOS-Chem and CAM-chem in CESM2. While both models show similar ozone budgets and concentrations within 5 ppb, they differ in key processes like photolysis schemes, aerosol effects, and halogen chemistry. Evaluation against observations suggests successful simulation of oxidants under polluted conditions but highlights potential biases in boundary layer mixing in CESM2. This integration supports the MUSICA vision of modularizing tropospheric chemistry in Earth system models.*

*Main comments*

1. *There are numerous global-scale atmospheric chemistry transport models available today. Could you provide more context on why selected these two particular models for comparison? Giving a more in-depth discussion on the scientific significance behind this choice would be helpful.*

We appreciate the reviewer's positive comments for our manuscript. As the reviewer notes, this work is motivated by the MUSICA vision of modularizing tropospheric chemistry and the emerging capability to explore the individual effects of different processes affecting chemistry in a common Earth System Model. We have clarified the context of our intercomparison of GEOS-Chem and CAM-chem and the choice of the two models in the introduction.

**Page 2, Section 1**

Here we compare two state-of-science atmospheric chemistry modules, GEOS-Chem and CAM-chem, within the Community Earth System Model (CESM2) (Danabasoglu et al., 2020). CAM-chem is the resident atmospheric chemistry module in CESM2 *and as such has a large user base*

(Lamarque et al., 2012; Tilmes et al., 2015, 2016; Emmons et al., 2020). GEOS-Chem is *used by hundreds of research groups worldwide* as an offline chemical transport model (CTM) driven by the GEOS archive of external meteorological data (Bey et al., 2001).

**Page 3, Section 1**

We analyze the individual processes driving differences between GEOS-Chem and CAM-chem, and use observations to arbitrate when possible. *This process-based intercomparison of GEOS-Chem and CAM-chem leverages the unique capability of comparing these two major representations side-by-side in a common ESM environment where specific causes of model differences can be attributed to different representations of chemistry.*

2. *This study has extensively compared vertical profiles, but what about comparisons of surface observational elements? My suggestion would be to include data from ground monitoring stations to assess ozone and nitrogen oxides.*

3. *In figure 4, I am wondering why surface ozone were rather high in western China (especially in regions like Tibet, almost the highest around the world) where anthropogenic emissions, i.e, NOx, were relatively low. Did you compare the surface simulations with surface observations?*

Thank you for your suggestion. We now include comparisons with surface ozone observations from background sites from the NOAA ESRL Global Monitoring Division (10 sites) and WMO-GAW sites in China (5 sites). Background sites were chosen due to the coarse resolution of the model and because only monthly average outputs were archived. Ozone is high over Tibet

because of its high elevation and therefore more representative of the free troposphere, with greater influence from the stratosphere, than the surface.

**Page 16, Section 5**

Figure 5 compares annual mean ozone vertical profiles simulated by GEOS-Chem and CAM-chem to ozonesonde observations for 2016 from the World Ozone and Ultraviolet Radiation Data Centre (WOUDC), averaged across 9 regions following Tilmes et al. (2012). *Figure 6 compares April 2016 monthly mean surface ozone simulated by GEOS-Chem and CAM-chem to background surface ozone observations from 10 sites of the NOAA ESRL Global Monitoring Division (McClure-Begley et al., 2013) and 5 remote sites in China under the World Meteorological Organization Global Atmosphere Watch Programme*. Both models match the observations well and are within 5-10 ppb of each other. GEOS-Chem has lower ozone at high northern latitudes *(up to 10 ppb at the surface)* because of halogen chemistry.

[Figure]

4. *Have these two models taken into account the impact of halogen chemistry mechanisms on the formation of photochemical ozone? If they have, please provide some discussion.*

Thank you for your question. GEOS-Chem incorporates tropospheric halogen chemistry and thus effects on reducing formation, and increasing destruction of ozone (Wang et al., 2021). CAM-chem does not explicitly represent tropospheric halogen chemistry and thus does not account for this effect. We have clarified this difference and its effects in the text.

**Page 10, Section 4**

Figure 4 shows annual mean surface and 500 hPa ozone and NOx concentrations simulated by GEOS-Chem and differences with CAM-chem. Ozone differences are generally smaller than 5 ppb, indicating a remarkable degree of agreement. The largest surface differences are at southern

mid-latitudes due to slower ozone deposition to the ocean in GEOS-Chem. At 500 hPa, GEOS-Chem has lower ozone at high latitudes due to tropospheric halogen chemistry. *This chemistry increases ozone destruction through catalytic ozone loss cycles driven by iodine and bromine, and decreases ozone production by conversion of NOx to halogen nitrates. Tropospheric halogen chemistry is not represented in the default configuration of CAM-chem*.

*Minor suggestions*

 1. **Line 155-160 "Fast-JX includes aerosol extinction but TUV does not, which explains the larger differences over polluted and open fire regions" Please give some examples to indicate these regions**

Thank you for the suggestion, we have clarified this in the text.

**Page 7, Section 3**. There are some larger differences in polluted and open fire regions such as in *East Asia and Siberia, and at high latitudes*.

 2. **Line 200-205, Please extend more about the recent update in isoprene oxidation chemistry why isoprene does not titrate OH in GEOS-Chem**

We have clarified this in the text.

**Page 10, Section 4**

Over the Amazon and Congo basins where NOx is low, isoprene does not titrate OH in GEOS-Chem due to recent updates in isoprene oxidation *chemistry incorporating H-shift isomerization of isoprene-hydroxy-peroxy radicals to recycle OH, which sustains OH under low-NO conditions* (Bates and Jacob, 2019).

*3. Why GEOS-Chem simulated NOx in oceans were notably higher than CAM-chem?*

We have clarified this in the text.

**Page 10, Section 4**

*Particulate nitrate photolysis in GEOS-Chem corrects for a missing NOx source in the remote troposphere (Shah et al., 2023) and accounts for the higher NOx and ozone than CAM-chem over the oceans.*

*4. Figure 5 uses pressure while figure 6 and 7 use altitude (km) to show height, it is suggested to use the same unit of height, for instance, kilometers.*

Thank you, this has been fixed.

**Reviewer #2**

*Li et al. implemented the GEOS-Chem atmospheric chemistry module into the CAM-Chem and evaluated the difference in O3 and OH chemistry between model simulations. They also conducted the model sensitivity analysis and discussed the difference in O3, NOx, CO, and OH due to underlying processes including the photolysis scheme, aerosol nitrate photolysis, N2O5 cloud uptake, halogen chemistry, and ozone deposition to the oceans. I appreciate the tremendous technical work involved in implementing the CESM-GC capacity. I suggest minor revisions before this paper is accepted in ACP.*

1. *The authors provided some high-level explanations of the underlying processes that lead to the difference across the model simulations. However, more detail is recommended. For instance, what is the process of scavenging water-soluble gases in convective updrafts? Are these gases NOx and less reactive VOCs? Do they mainly affect O3 formation in the upper troposphere?*

Thank you for your question. The process of scavenging soluble tracers in convective updrafts is represented in the standalone GEOS-Chem by Balkanski et al. (1993) and Liu et al. (2001), but not in CESM2 except for aerosols represented in MAM4. Thus, all other species, including gas-phase species and GEOS-Chem aerosols such as nitrate, are not scavenged and have an unphysical buildup in the upper troposphere. We have clarified this in the introduction.

**Page 3, Section 1**

CESM2 does not *couple convective transport with* scavenging of water-soluble species, but this is a major process in the GEOS-Chem CTM to prevent unphysical buildup of *water*-soluble species in the upper troposphere (Balkanski et al., 1993; Liu et al., 2001). *If convective transport*

*and scavenging are applied sequentially, instead of being coupled, then water-soluble species can reach the upper troposphere in deep convective updrafts and disperse on the model grid scale to avoid scavenging.* Indeed, Fritz et al. (2022) found large overestimates of upper tropospheric aerosol in GEOS-Chem within CESM2 as compared to the offline GEOS-Chem.

2. ***Could you provide some discussion on the halogen chemistry mechanism implemented in the model? How large is the uncertainty in this mechanism? Is the uncertainty introduced through the chemical mechanism smaller than the difference observed here with and without halogen chemistry?***

Thank you for your question. Previous implementations in the offline GEOS-Chem model (Sherwen et al., 2016b; Wang et al., 2021) and CAM-chem (Saiz-Lopez et al., 2012) have shown tropospheric ozone to be 10-19% lower with halogen chemistry. While the magnitude of the effect is uncertain and model-dependent, all model implementations show a reduction in tropospheric ozone, which we also observe in GEOS-Chem within CESM2.

**Page 11, Section 4**

*The magnitude of the effect of halogen chemistry on ozone is uncertain, ranging from 10% to 19% in previous implementations in offline GEOS-Chem (Sherwen et al., 2016b; Wang et al., 2021) and CAM-chem (Saiz-Lopez et al., 2012), but all models report lower tropospheric ozone as a result. We find that halogen chemistry has a smaller effect on tropospheric ozone in GEOS-Chem within CESM2 than offline* due to weaker wind speeds and lower sea surface temperatures in CESM2, resulting in weaker sea salt and gaseous iodine emissions.

*3. For the calculation of OH in Table 2, is it air mass-weighted column OH?*

Yes, this is global annual mean air-mass-weighted OH in the troposphere.

*4. In Table 2, both models generate the same OH. However, from Figure 3, the difference in OH is considerably large, with -3.2% at the surface and -10.1% at 500 hPa. Could you explain this discrepancy?*

Thank you for your question. Table 2 is calculated with the air mass-weighted tropospheric OH (in units of molecules cm$^{-3}$) but Figure 3 is calculated using mixing ratios (in units of parts-per-quadrillion). The difference in tropospheric OH mass (non-mass-weighted) is -3.2% in GEOS-Chem (0.224 Gg) compared to CAM-chem (0.231 Gg), consistent with Figure 3.

We have removed the inset differences in the Figures to avoid confusion.

*5. The spatial resolutions of both models are coarse, how does the model's spatial resolution affect the model comparison against observations, especially those from ATOM1 and KORUS-AQ observations?*

Thank you for your question. The coarse resolution of the model primarily affects comparisons in the boundary layer as the model cannot resolve fine spatial scales, but comparisons with aircraft observations above the boundary layer should be less affected as species are more well mixed horizontally at altitude with faster wind speeds. Jo et al. (2023) has shown that differences between resolutions are smaller at altitude; previous comparisons of CAM-chem with KORUS-AQ such as Gaubert et al. (2020) have also used 0.9° × 1.25° resolution as in this work. The resolution for this work is also well adapted to the scales sampled by ATom. We have clarified in text how the model is sampled along aircraft tracks.

**Page 16, Section 5**

*Model profiles compared to aircraft observations are computed at model runtime by sampling the 2 closest timesteps and 4 closest grid boxes to the time-varying flight track data, and then interpolated to the aircraft time and location. The $0.9^o \times 1.25^o$ resolution of the simulation is well adapted to the scales sampled by ATom.*

6. ***Both models consistently show a high bias in O3 compared to observations, except for GEOS-Chem without the PNO3 photolysis. Could you discuss more on the possible causes of the high O3 bias? For instance, how do stratospheric O3 and lightning NOx emissions contribute to O3 in the upper troposphere?***

Thank you for your question. Other studies using CESM/CAM-chem have investigated the high ozone bias in the model and found effects related to horizontal resolution, which would affect artificial dilution of emissions, and parameterizations such as lightning $NO_x$ and biogenic emissions, as well as meteorology (Schwantes et al., 2022; Jo et al., 2023). The cause of ozone bias and differences between models will be addressed by future work, which will include simulations at higher resolutions.

**Page 20, Section 6**

*Both models are too high compared to observations above 4 km altitude, which is due at least in GEOS-Chem to excessive particulate nitrate resulting from inadequate convective scavenging.*

**Page 23, Section 7**

*Comparison with KORUS-AQ aircraft observations allowed model evaluation for polluted conditions. Ozone concentrations in GEOS-Chem and CAM-chem are higher than observed*

*above 4 km altitude, which in GEOS-Chem is due to excessive particulate nitrate photolyzing to produce excessive NOx.*

**7. The figures are vague, please update them to a higher resolution.**

Thank you for the suggestion, we have increased the resolution of the figures in text. We will have also uploaded full-resolution figures in the production and typesetting process.

**Reviewer #3**

*My apologies for this late review. I congratulate the Editor for pushing this paper along. The paper 2024-470 by Haipeng Lin et al is a very well written study that examines the tropospheric ozone budget and sensitivities for two very different chemistry modules (GEOS-Chem & CAM-chem) operating within a common framework (CESM2). The writing is incredible; I did not find any typos or very awkward sections! (Well except for BCC?) There are a few minor fixes and one more substantive issue that might be fixed before publication. L27ff: this list of 5 specific items are not treated equally or evaluated equally well in the paper and hence do not belong so strongly as in the abstract. e.g., N2O5 cloud uptake and oceanic ozone deposition.*

Thank you for the suggestion. We believe these factors still should be included in the abstract to provide clarity in the differences in the two models, as shown in the budget term analysis in Table 3, as well as important factors in interpreting differences in $NO_x$ and ozone shown in Section 4.

*L30ff. Yes, CAM-chem is notorious for being very low in CO compared with many models and it seems to be something deeper than attributed here simply to isoprene or OH.*

Thank you. We have revised to note that there may be other factors affecting low CO in CAM-chem which is out of scope for our manuscript.

**Abstract**. Carbon monoxide is lower in CAM-chem (and lower than observations*), at least in part because* of higher OH concentrations in the northern hemisphere and insufficient production from isoprene oxidation in the southern hemisphere.

*L39: "nonlinear" is used too easily in our community and often incorrectly.  It is also meaningless since most of these processes are quasi-linear.  how about: "by second-order processes that couple across hydrogen…."*

Thank you for your suggestion. We have revised the wording.

**Page 2, Section 1**.

It is produced within the troposphere by *complicated chemical mechanisms* involving hydrogen oxide radicals (HOx ≡ OH + peroxy), nitrogen oxide radicals (NOx ≡ NO + NO2), volatile organic compounds (VOCs), and ozone itself.

*L41:  It is "chemistry-transport models (CTMs) NOT chemical-transport since these models do more than just transport the chemicals, they are chemistry models.  You got it right with chemistry-climate models so fix this please.*

*L53 "chemistry is coupled to transport"  What do you mean here?  it is always, unless you are talking about models that are NOT operator split?  i.e. simultaneous chemical rates and transport?  If you mean that the resulting chemicals can change the dynamics through radiation then this is a CCM, OK.  BUT you model here is forced with MERRA-2 fields and hence is NOT a CCM.  Overall the distinction between offline and online is becoming blurred*

*and your use of it here is not helping. If you run within CESM with forced met fields then it is not "online", it is the same as a CTM that you call offline. I really recommend you drop off/online and come up with a simpler expression. Anyway all the runs here are the same so why bother. Are you running as a full CCM forced by GHGases and SSTs in all these cases?*

Thank you for the suggestions. Brasseur and Jacob (2017) define CTMs as *chemical transport models*, which includes both 3-D transport *and chemical evolution* of the atmosphere. Other established literature on modeling (e.g., Baklanov et al., 2014) use the same terminology. The model here is nudged using MERRA-2 fields (with a 50-hr relaxation time) with an online land model. The model is forced by GHGs and SSTs in all cases. In our model configuration meteorology is simulated at the same time as chemistry, and feedback from chemistry is possible, thus it is an "online" model, in contrast to an "offline" model like standalone GEOS-Chem where the model is fully driven by meteorological archives.

For clarity, we have clarified the definition of "online" and "offline" models in text and the model configuration used in this work.

**Page 2, Section 1**.

GEOS-Chem is used by hundreds of research groups worldwide as an offline chemical transport model (CTM) driven by the GEOS archive of external meteorological data (Bey et al., 2001). *Offline here is defined by contrast to online models that perform their own simulations of atmospheric dynamics (Brasseur and Jacob, 2017).* GEOS-Chem is grid-independent and modularized, so that the chemical module describing local operations in 1-D model columns (including emissions, chemistry, and deposition) is separated from the transport module (Long et al., 2015). This allows independent implementation of the GEOS-Chem chemical module in *online* models, *where chemical transport is done as part of the simulation of atmospheric*

*dynamics (Hu et al., 2018; Lin et al., 2020; Lu et al., 2020; Keller et al., 2021). The GEOS-Chem*

chemical module has been previously coupled to the WRF and GEOS meteorological models to

investigate aerosol-chemistry-climate feedbacks (Feng et al., 2021; Moch et al., 2022) *and*

*powers the GEOS global chemical forecasts (GEOS-CF) (Keller et al., 2021).* The same *GEOS-*

*Chem* scientific code base is used *as* in the offline CTM such that version updates developed for

the CTM can be seamlessly passed on to the online applications.

**Page 3-4, Section 2.1**.

*We use the "F" compsets in CESM which use active atmosphere and land models with*

*prescribed sea surface temperatures, sea ice, and greenhouse gases for current climate (CMIP6*

*SSP2-4.5 scenario). The model reproduces a given meteorological year by nudging winds and*

*temperature* (using the "FCnudged" configuration in CAM6) *to* 3-hourly MERRA2

*meteorological reanalysis produced by* the NASA Global Modeling and Assimilation Office.

*This nudging is done with a 50-hour relaxation time that allows CAM to generate its own*

*physics, including the hydrological cycle and the effects of aerosols on clouds.*

***L65ff: This large number -- 10-30% lower trop O3 from Br reactions -- seems like it is over***

***exaggerated and should not be repeated since later you say that this was due to a major***

***mistake with iodine, not bromine?***

Thank you for the suggestion. We removed the percentage and revised it to clarify the difference

seen is due to tropospheric halogen chemistry.

**Page 3, Section 1**.

They found good agreement between the two modules for stratospheric ozone, but lower tropospheric ozone in GEOS-Chem due to *tropospheric halogen chemistry* not considered in CAM-chem.

***L115ff – again the use of offline v. online is only confusing here.***

We have revised throughout to ensure clear distinction between the online CESM2 configuration of GEOS-Chem and the offline, CTM configuration.

***L227: how can a modern CCM like CESM2 have SSTs that are so far off from observed???***

Thank you for the question. This is due to 2M temperature being incorrectly used as SST in the interface to HEMCO and is not an issue in CESM2. We have clarified this in text.

**Page 11, Section 4**.

Fritz et al. (2022) previously found tropospheric ozone in GEOS-Chem to be 30% lower than CAM-chem in the extratropics because of halogen chemistry, *but iodine emissions in that simulation were 100-fold too high because the interface to HEMCO erroneously passed 2-meter temperature instead of sea surface temperature to the iodine emissions module* (Section 2.2). With corrected iodine emissions we find only a 4% decrease of tropospheric ozone in GEOS-Chem due to tropospheric halogen chemistry.

***L230ff – Table 2 & Table 3: This gets very interesting. Looking hard at these tables, I am beginning to believe that CESM2 simply does not conserve O3. Is this so? Surely you have some diagnostics for this? I have heard talks saying that CESM loses 200 Tg/y of trop O3 but not seen it documented.***

Thank you for the observation. The $d(O3_{trop})/dt$ in GEOS-Chem and CAM-chem simulations in this work is +0.4 Tg a$^{-1}$ and -4.9 Tg a$^{-1}$ which is small compared to the tropospheric ozone burden of 350 Tg and 342 Tg, respectively. We have added it to the Table 2 footnote.

**Page 12, Table 2 Footnote**. *c. Residual of mass balance between tropospheric chemical production and loss, Ox deposition, and accumulation. This term represents an estimate of stratosphere-troposphere exchange (STE) in the absence of advective flux diagnostics in CESM2. The accumulation term in the GEOS-Chem and CAM-chem models over 2016 is 0.4 Tg a-1 and -4.9 Tg a-1, respectively.*

**The idea of using a residual to get STE fluxes is very old and should not be propagated here. Since there is not evidence that you can calculate the STE flux, questionable evidence that you conserve O3, and obvious evidence that POx-LOx is only good to 5-10% (see Prather Elementa paper on the POX & LOX not being the same as dO3/dt, doi: 10.1525/elementa.2023.00112). You should not presume the accuracy of all you terms by declaring the residual to be STE. It does not matter that this is traditional practice, at least call it unknown residual, but give us an uncertainty in the terms here.**

Thank you, we have changed the table term to "$O_x$ residual term (including STE)" and revised the footnote. The residual term represents our best estimate of STE given that there is no advective flux diagnostic in CESM2 which would provide a more accurate diagnostic of ozone transport across the tropopause.

**Page 11, Section 4.**

*Table 2 includes a residual term in the tropospheric ozone budget as a balance between the chemical production, chemical loss, deposition, and (negligible) accumulation terms. This*

*residual term of 341-380 Tg a-1 is expected to represent stratosphere-troposphere exchange*

*(STE), which is not explicitly diagnosed in CESM2, and falls within the range of literature values*

*listed in Table 2.*

**The Table 3 sensitivity results raise some interesting questions – the P-L does not change with**

**'J-nitrate', but it increases by 100 Tg/y for 'no-N2O5-uptake'.  Why?**

**Likewise, the Fast-JX in CAM-chem increases P-L from 587 to 764 Tg/y – where does that**

**excess O3 go? to the surface? STE should not change, or does the stratosphere also change**

**with Fast-JX?  You should document that here.**

Thank you for the question. In both runs ozone increases throughout the troposphere, including

the surface. Surface ozone is 2.1 ppb higher on average in CAM-chem with Fast-JX. An increase

in tropospheric ozone would increase deposition and decrease STE, which is consistent as a

change in P-L is mirrored by a change in D-STE.

**Page 11, Section 4**.

*The residual changes slightly in the sensitivity simulations of Table 3 in a way that is consistent*

*with the tropospheric ozone burden, as increasing tropospheric ozone decreases STE while*

*increasing deposition.*

**L258 & Fig 6: for comparing with profiles of OH and CO made by aircraft, you must simulate**

**the correct time of day.  How did you take these values form the model?  Did you actually pull**

**off the nearest hour to when the obs were made?  Is the mean profile correctly weighted?**

**Very tough to do this with a CCM, I am impressed.  Just note that your flight-track sampling**

**included the correct local solar time ! (L290).**

*By the way, producing a mean profile when all the points that went into it were sampled most likely in a systematic, weirdly biased time-of-day makes this a pretty much useless diagnostic for anyone else.*

Thank you for the question. Model results for comparison with aircraft observations are computed at model runtime along the time-varying flight track information, which specifies the time, latitude, and longitudes of the desired track. During the model run, the model is sampled at the 2 nearest timesteps and 4 nearest grid-boxes, interpolated, and the result is output. The CAM-chem Wiki has further details

(https://wiki.ucar.edu/display/camchem/Changing+Output%3A+Frequency%2C+Species+and+Regions); this approach is used consistently with other studies using CAM-chem. We have clarified this in the text.

**Page 16, Section 5**

*Model profiles compared to aircraft observations are computed at model runtime by sampling the 2 closest timesteps and 4 closest grid boxes to the time-varying flight track data, and then interpolated to the aircraft time and location.*

*L312: typo? what is a BCC ESM, spell out.*

Thank you for noticing. This is the Beijing Climate Center (BCC) Earth System Model which has also been coupled to GEOS-Chem (Lu et al, 2020). We have spelled it out.

**Page 20, Section 6**.

A solution would be to replace CESM convective transport with the GEOS-Chem offline convective transport and scavenging module using archived CESM convective mass fluxes, and

this has been done before when coupling GEOS-Chem to the GEOS and *Beijing Climate Center (BCC) ESMs* which had the same problem of not scavenging water-soluble species in convective updrafts (Yu et al., 2018; Lu et al., 2020).

References:

Baklanov, A., Schlünzen, K., Suppan, P., Baldasano, J., Brunner, D., Aksoyoglu, S., Carmichael, G., Douros, J., Flemming, J., Forkel, R., Galmarini, S., Gauss, M., Grell, G., Hirtl, M., Joffre, S., Jorba, O., Kaas, E., Kaasik, M., Kallos, G., Kong, X., Korsholm, U., Kurganskiy, A., Kushta, J., Lohmann, U., Mahura, A., Manders-Groot, A., Maurizi, A., Moussiopoulos, N., Rao, S. T., Savage, N., Seigneur, C., Sokhi, R. S., Solazzo, E., Solomos, S., Sørensen, B., Tsegas, G., Vignati, E., Vogel, B., and Zhang, Y.: Online coupled regional meteorology chemistry models in Europe: current status and prospects, Atmos. Chem. Phys., 14, 317–398, https://doi.org/10.5194/acp-14-317-2014, 2014.

Gaubert, B., Emmons, L. K., Raeder, K., Tilmes, S., Miyazaki, K., Arellano Jr., A. F., Elguindi, N., Granier, C., Tang, W., Barré, J., Worden, H. M., Buchholz, R. R., Edwards, D. P., Franke, P., Anderson, J. L., Saunois, M., Schroeder, J., Woo, J.-H., Simpson, I. J., Blake, D. R., Meinardi, S., Wennberg, P. O., Crounse, J., Teng, A., Kim, M., Dickerson, R. R., He, H., Ren, X., Pusede, S. E., and Diskin, G. S.: Correcting model biases of CO in East Asia: impact on oxidant distributions during KORUS-AQ, Atmos. Chem. Phys., 20, 14617–14647, https://doi.org/10.5194/acp-20-14617-2020, 2020.

Jo, D. S., Emmons, L. K., Callaghan, P., Tilmes, S., Woo, J. H., Kim, Y., Kim, J., Granier, C., Soulié, A., Doumbia, T., and Darras, S.: Comparison of Urban Air Quality Simulations During the KORUS-AQ Campaign With Regionally Refined Versus Global Uniform Grids in the Multi-

Scale Infrastructure for Chemistry and Aerosols (MUSICA) Version 0, J. Adv. Model. Earth Sy., 15, e2022MS003458, https://doi.org/10.1029/2022MS003458, 2023.

Schwantes, R. H., Lacey, F. G., Tilmes, S., Emmons, L. K., Lauritzen, P. H., Walters, S., Callaghan, P., Zarzycki, C. M., Barth, M. C., Jo, D. S., Bacmeister, J. T., Neale, R. B., Vitt, F., Kluzek, E., Roozitalab, B., Hall, S. R., Ullmann, K., Warneke, C., Peischl, J., Pollack, I. B., Flocke, F., Wolfe, G. M., Hanisco, T. F., Keutsch, F. N., Kaiser, J., Bui, T. P. V., Jimenez, J. L., Campuzano-Jost, P., Apel, E. C., Hornbrook, R. S., Hills, A. J., Yuan, B., and Wisthaler, A.: Evaluating the Impact of Chemical Complexity and Horizontal Resolution on Tropospheric Ozone Over the Conterminous US With a Global Variable Resolution Chemistry Model, J. Adv. Model. Earth Syst., 14, e2021MS002889, https://doi.org/10.1029/2021MS002889, 2022.